# Field evaluation of thermal and acoustical comfort in eight North-American buildings using embedded radiant systems

**Megan Dawe** [1]*, **Caroline Karmann**[2], **Stefano Schiavon** [1], **Fred Bauman**[1]

**1** Center for the Built Environment (CBE), University of California, Berkeley, California, United States of America, **2** Laboratory of Integrated Performance in Design (LIPID), École Polytechnique Fédérale de Lausanne (EPFL), Lausanne, Switzerland

* megan.dawe@berkeley.edu

**Data Availability Statement:** All relevant data are within the manuscript and its Supporting information file. The full dataset cannot be shared publicly due to the nature of the Institutional Board

## Abstract

We performed a post-occupancy assessment based on 500 occupant surveys in eight buildings using embedded radiant heating and cooling systems. This study follows-up on a quantitative assessment of 60 office buildings that found radiant and all-air buildings have comparable temperature and acoustic satisfaction with a tendency for increased temperature satisfaction in radiant buildings. Our objective was to investigate reasons of comfort and discomfort in the radiant buildings, and to relate these to building characteristics and operations strategies. The primary sources of thermal discomfort are lack of control over the thermal environment (both temperature and air movement) and slow system response, both of which were seen to be alleviated with fast-response adaptive opportunities such as operable windows and personal fans. There was no optimal radiant design or operation that maximized thermal comfort, and building operators were pleased with reduced repair and maintenance associated with radiant systems compared to all-air systems. Occupants reported low satisfaction with acoustics. This was primarily due to sound privacy issues in open-plan offices which may be exacerbated by highly reflective surfaces common in radiant spaces.

## 1 Introduction

Energy consumption in US commercial buildings accounts for 18% of the country's primary energy use, with 30% of that being HVAC related [1]. There is a need to reduce buildings' energy use to achieve carbon emission goals. In parallel, the building industry is becoming increasingly aware that the indoor environment impacts our health and well-being. A typical person spends around 90% of their lives indoors [2]. This long exposure to indoor environments pushes us to rethink the design and operation of our most common spaces in order to address and support occupants' well-being, performance, and health. Researchers and building professionals seek design strategies to simultaneously address the dual challenges of indoor environmental quality (IEQ) and energy use.

Review at the University of California, Berkeley approval, which specifies that only research staff at the Center for the Built Environment may access the data. Researches cannot apply for IRB approval to access the data. We have provided the minimum data set as defined by PLOS ONE, which includes data relevant for this study in aggregate for each building and allows for replication of results and figures.

**Funding:** This work was funded by the Center for the Built Environment at University of California Berkeley. The Center for the Built Environment, with which the authors are affiliated, is advised by and funded in part by many partners that represent a diversity of organizations from the building industry – including manufacturers, building owners, facility managers, contractors, architects, engineers, government agencies, and utilities. None of the advisors participated directly in the research.

**Competing interests:** The authors have declared that no competing interests exist.

Radiant heating and cooling systems are thermally controlled surfaces that exchange heat mainly through thermal radiation. Radiant systems are relatively new to North America, and an NBI study found Zero Net Energy commercial buildings often use radiant systems [3]. Within the larger family of radiant systems, embedded radiant systems such as thermally activated building systems (TABS) and embedded surface systems (ESS) operate at relatively low-temperature for heating and high-temperature for cooling. These systems have the potential to achieve significant energy savings [4]. Radiant buildings also have to meet the occupants' needs for comfort and workspace quality. These objectives often remain difficult to address in the day-to-day operation of commercial buildings, primarily due to the limited understanding of human comfort and its in situ assessment.

Radiant systems are commonly assumed to provide improved thermal comfort in comparison to all-air systems. Cited theories for improved thermal comfort include creating uniform thermal conditions in a space [4,5], reducing risk of unwanted air movement [6–9], and reducing or eliminating discomfort from hot or cold surfaces (i.e., radiant asymmetry) [9,10]. Karmann et al. completed a critical literature review to learn if spaces using radiant system provide better, worse, or similar thermal comfort compared to spaces using an all-air system [11]. Their review revealed a lack of studies based on occupant's perception, while more studies relied on calculated thermal comfort. Considering the limited number of studies available and the small sample sizes for each study, their review could not establish a definitive statement on the effectiveness of radiant systems for thermal comfort. Aside from thermal comfort, little has been studied about the ways in which radiant systems affect space acoustics. Radiant systems are commonly installed on large acoustically reflective surfaces (e.g., ceilings or floors) that are kept exposed to maximize thermal radiation. In practice, exposed concrete surfaces can lead to increased reverberation and lower acoustic satisfaction, but this assumption requires validation from the field considering spaces experienced by building occupants.

## 2 Background

Karmann et al. found few existing studies used occupant surveys to compare comfort in spaces with radiant systems to those with all-air systems [11]; therefore, Karmann et al. conducted a quantitative survey study to determine whether radiant systems provide higher satisfaction than all-air systems [12]. The study included 26 radiant buildings (1,645 occupant responses) and 34 all-air buildings (2,247 occupant responses) of comparable key characteristics (e.g., building size, year built, climate zones). 21 of the 26 radiant buildings (over 2/3 of the individual responses) used TABS or ESS. Fig 1 shows boxplots for occupant satisfaction by conditioning type. Temperature satisfaction is the only category that showed a difference in median between the two subsets. The difference in mean was statistically significant (p<0.001), and the Spearman's $\rho$ effect size of the difference ($\rho = 0.14$) was the largest observed in this study; however, the practical difference shall be considered as either negligible or small [13,14]. Karmann et al. [12] concluded that indoor environmental quality is the same with a tendency for increased thermal satisfaction in radiant buildings. Acoustical categories ranked as the lowest performing for both systems, and neither noise nor sound privacy satisfaction showed statistical or practical significant differences between the two subsets. A mixed effects model showed that 21% of the variance for sound privacy could be described by 'between office type' differences (e.g., private, open-plan, etc.), which is more than the variance explained by conditioning systems.

## 3 Objective

The quantitative analysis from (Karmann et al. 2017) was able to show trends and provide answers to the commonly asked questions comparing radiant to all-air systems. Yet, it was

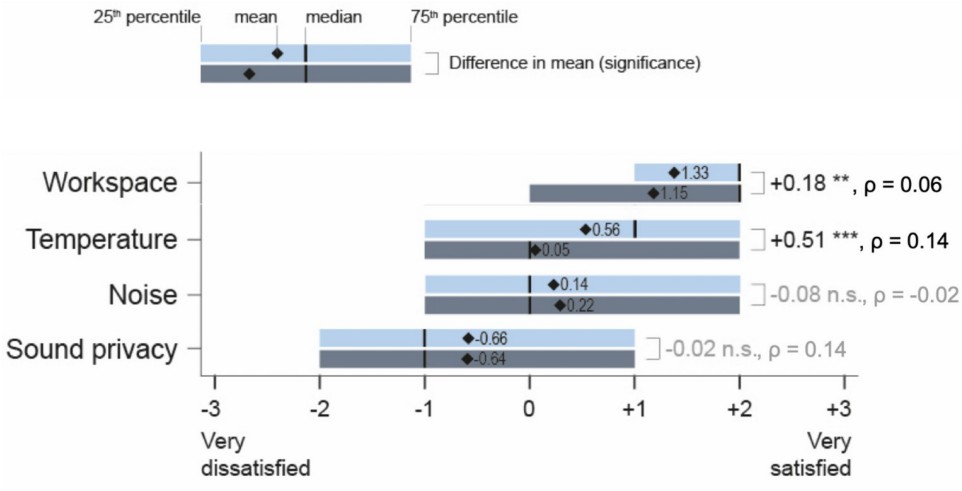

**Fig 1. Boxplot for occupant satisfaction with workspace, temperature and acoustics in 60 radiant and all-air buildings.** The differences in mean values (radiant–all-air) statistical and practical significance (effect size) of this difference are indicated on the right.

grounded only on aggregated answers to a post-occupancy survey and was detached from the idiosyncratic characteristics of each building. Further, the majority of thermal comfort evaluations in radiant spaces are in laboratory conditions or small-scale field studies, and reported comfort is based on calculated PMV-PPD which has very low prediction accuracy [15]. To expand on previous assessments, we selected eight TABS or ESS radiant buildings to further investigate reasons of thermal and acoustical satisfaction, and to uncover relations, if any, between occupants' impressions to building characteristics and operation strategies. Our post-occupancy evaluation approach was to find shared themes across several buildings rather than to focus on the specific and unique outcomes of each individual building. We accomplished this through in-depth review of over 500 occupant surveys (including open-ended comments) and interviews of building operators to provide useful lessons learned and insight into occupant satisfaction in buildings using radiant systems.

With this study, we want to provide insight into occupant satisfaction and perception that can help designers and building operators address or plan for improved occupant comfort and well-being in radiant buildings.

## 4 Methods

### 4.1 Building selection

For this study, we targeted office-type buildings located in different climates in North America and using embedded types of radiant system (i.e., TABS or ESS) for which occupant satisfaction surveys, building characteristic, and radiant design data was available or could be acquired. All buildings use the radiant system for cooling and heating. In buildings where only select spaces use radiant, we limited occupant surveys to those spaces. We selected these building systems because embedded systems have a much longer time constant than panel systems [16], and may have thermal comfort and acoustical issues. Moreover, we expect they would have the possibility of thermal storage (grid to building load flexibility), improved energy performance, and lower cost than radiant panel buildings. We intentionally included an array of

buildings performing well or poorly in either or both occupant satisfaction or energy performance to better assess sources affecting comfort and the possible correlation to energy performance.

## 4.2 Occupant satisfaction survey

This study relied on occupant satisfaction data collected by the Center for the Built Environment (CBE) at the University of California, Berkeley [17,18]. The IEQ occupant satisfaction survey covers nine core categories including thermal comfort and acoustics [19]. It uses a 7-point Likert scale ranging from 'very satisfied' (+3) to 'very dissatisfied' (-3), with a 'neutral/neither satisfied nor dissatisfied' midpoint (0). Dissatisfied responses trigger branching questions targeting the source of the dissatisfaction. The survey does not include similar branching questions when an occupant expresses satisfaction; this was done as a method to minimize survey fatigue. The voluntary and web-based survey also includes fields for open-ended responses. The surveys are intended to gather long-term occupant experiences rather than "right now" surveys which might be paired with objective information such as environmental measurements and current system operation.

The analysis uses the survey's branching questions and open-ended responses to identify sources of satisfaction and dissatisfaction. The open-ended responses allowed us to also infer sources of satisfaction or dissatisfaction that were not captured within the branching questions. We reported some of these comments (as quotes) when they captured interesting insights about comfort conditions experienced in their space. We calculated the percentage of dissatisfied occupants across all buildings and per building for both temperature and acoustic satisfaction, considering the different satisfaction intervals described in standards and common definitions. The survey response rate depends on the willingness of occupants to provide feedback. Therefore, we did not establish a threshold for minimum response rate. We reported the response rate for each building and considered all occupant feedback as suggestive of trends, regardless of response rate achieved.

The Institutional Review Board at the University of California, Berkeley approved this work (IRB-2010-05-1550). We administered all surveys online and consent was informed; participants provided their written consent online prior to advancing to the survey. The study did not include minors. We used R v.3.5.0 (R Development Core Team 2017) for all numerical analysis.

## 4.3 Building and radiant system characteristics survey

For each building in this study, we also collected data on the building's characteristics, radiant design, and facility and system operations. Information collected includes type of radiant system, control strategy, temperature setpoints, zone sizes, ventilation strategy, window and shading design, presence of other HVAC systems, and more. The building manager, the facilities manager, or a member of the design team provided this information in an online survey. We used this information to detect any relationship between occupant satisfaction and building characteristics or radiant design.

The data collection did not include physical measurements or occupant tracking information. The analysis relies exclusively on anonymous survey responses and selected interviews.

## 4.4 Building operator interviews

We interviewed the primary engineer and/or knowledgeable contacts with the building operation team for six buildings; contacts at two buildings did not respond to our requests. Five of these interviews took place by telephone, and one was conducted during a site visit. The goals

of the interviews were to gain their perspective on: 1. occupant feedback for IEQ parameters; 2. the balance and synergies between energy performance and occupant thermal comfort; 3. lessons learned during commissioning and operation.

# 5 Results and discussion

The following sections provide aggregated findings based on occupant experiences in their spaces and suggestions for building designers and operators. Occupant satisfaction data is provided in S1 Dataset.

## 5.1 Description of the buildings

The eight buildings selected for the qualitative analysis use embedded radiant systems (either TABS or ESS) for both heating and cooling, are of varying sizes and design, and located in five different ASHRAE 90.1 climate zones in North America. Table 1 summarizes the building characteristics, including thermal comfort ranking, and Table 2 provides the HVAC, comfort, and energy concept for each building. As seen in Table 1, the buildings represent a range of occupant comfort and energy performance. Energy performance is categorized by annual Energy Use Intensity (EUI) which incorporates all energy sources and an ENERGY STAR Score, which normalizes energy use by key drivers, including building size, location, number of occupants, and operating hours, and number of computers. Buildings B4 and B5 performed the best in terms of thermal comfort satisfaction relative to the other six buildings. Interestingly, B2 and B6 were low performing buildings and had the highest annual Energy Use Intensity (EUI) by more than double the next closest building. All buildings are LEED certified, with many achieving Platinum certification. Four buildings were designed with supplemental cooling equipment, but it is unknown what portion of sensible and latent cooling these systems serve in operation and how that might impact thermal comfort.

**Table 1. Building characteristics.**

| Bldg. ID | Function | Building size (m²) | Year built (original)[a] | Certifications | Location | ASHRAE climate zone | IEQ thermal satisfaction rank[b] | EUI[c] (kWh/m²) | ENERGY STAR Score[d] |
|---|---|---|---|---|---|---|---|---|---|
| B1 | Office | 4,831 | 2003 | LEED Platinum, Living Building Challenge | Seattle, WA | Mixed-marine (4C) | 14th/26 | 38 | 99 |
| B2[e] | Library | <10,000 | ≤ 2010 (renovated) | LEED Gold | - | Mixed-marine (4C) | 26th/26 | 486 | 1 |
| B3[f] | Office + Multi-purpose | 18,859 | 2015 (1910) | LEED Platinum, LEED EBOM | San Diego, CA | Warm-dry (3B) | 17th/26 | *unknown* | *Unknown* |
| B4 | Office | 16,016 | 2015 | LEED Platinum, Net zero | Fremont, CA | Warm-marine (3C) | 3rd/26 | 75 | 100 |
| B5 | Office | 33,445 | 2010 | LEED Platinum | Golden, CO | Cool-dry (5B) | 9th/26 | 114 | 98 |
| B6 | Office | 4,088 | 2010 (1986) | LEED Platinum | Atlanta, GA | Warm-humid (3A) | 21st/26 | 555 | NA[g] |
| B7 | Office + Lab | 1,512 | 2012 | LEED Platinum | Victoria, BC | Mixed-marine (4C) | 12th/26 | 151 | 98 |
| B8 | Office + Multi-purpose | 18,581 | 2012 | LEED Platinum | Sacramento, CA | Warm-dry (3B) | 16th/26 | *unknown* | *unknown* |

[a] In case the building was renovated, we indicated original year of construction in parenthesis.

[b] We ranked our buildings based on mean temperature satisfaction out of the 26 radiant buildings in the study [12].

[c] EUI: Annual Energy Use Intensity inclusive of all energy sources(from [20], converted to kWh/m²).

[d] ENERGY STAR Score yields a 1-to-100 percentile ranking, from [20].

[e] Building B2 requested to be anonymous (non-trackable), therefore we did not provide identifying information.

[f] Building B3 was ranked in IEQ thermal performance using the office portion only.

[g] Buildings must be at least 5,000 square feet to calculate an ENERGY STAR Score.

**Table 2. Comfort and energy concept of the building.**

| Bldg. ID | Radiant type[a] | Radiant zone portion[b] | Ventilation type[c,d] | Ventilation distribution[c] | Supplemental Cooling System | Htg/Clg Setpoints | System operation | Unoccupied operation[c,e] | Adaptive opportunities[f] | Acoustic treatment | Shading[c,g] |
|---|---|---|---|---|---|---|---|---|---|---|---|
| B1 | ESS (floor) | 80% | MM (change-over) | Overhead (DOAS) | None | 20/26 °C | Constant flow, variable temperature | 24/7 with setback | Operable windows, ceiling fans | *unknown* | i(o),e(o) |
| B2 | TABS (ceiling) | 100% | MM (change-over) | Underfloor (DOAS) | Upsized DOAS | 23/26 °C | Constant flow, variable temperature | 24/7 without setback | Operable windows, desk fans* | Carpet | e(f) |
| B3 | ESS (floor) | 40% | MM (unknown) | Overhead (DOAS) | None | 21/24 °C | Variable flow, variable temperature | 24/7 with setback | Operable windows, desk fans*, ceiling fans | *unknown* | i(o) |
| B4 | TABS (floor) | *51–75%* | MM (concurrent, change-over) | Overhead (DOAS) | Active chilled beams | 20/23 °C | Variable flow, variable temperature | Turns on before occupancy | Operable windows, desk fans*, heaters*, thermostat | Carpet, wall panels | i(o) |
| B5 | TABS (ceiling) | 100% | MM (unknown) | Underfloor (DOAS) | Fan coils and upsized DOAS | 22/26 °C | Variable flow, constant temperature | Turns on before occupancy | Operable windows, desk fans*, ceiling fans | Carpet, tall partitions, white noise generator | e(f) |
| B6 | ESS (ceiling) | *76–100%* | MV (fully) | Underfloor (DOAS) | *Unknown* | 21/23 °C | *unknown* | *unknown* | Desk fans*, heaters* | Carpet | i(o) |
| B7 | TABS (floor) | 100% | MM (change-over) | Trickle vent (DOAS) | Yes, only in conference rooms | 21/24 °C | Variable flow, constant temperature | 24/7 | Trickle vent, thermostat | VanAir doors[h] | i(o) |
| B8 | TABS (ceiling) | 100% | MV (fully) | Overhead (DOAS) | Considering adding heat pumps[i] | 21/24 °C | Variable flow, constant temperature | Turns on before occupancy | Desk fans*, heaters*, ceiling fans | Vertical ceiling panels | e(f) |

[a] Embedded surface systems (ESS), thermally activated building systems (TABS).

[b] Percent of building served by radiant system.

[c] Applies to the radiant zones of the building.

[d] MV: Mechanical ventilation (no operable windows), NV: Natural ventilation, MM: mixed-mode (type: change-over, concurrent, zoned).

[e] How the radiant system is operated during unoccupied hours.

[f] Adaptive opportunities may refer to fast-response actions that either affect groups (i.e., operable windows, ceiling fans) or individuals (i.e., desk fans, heaters). We used an asterisk to indicate opportunities supporting individual actions.

[g] Shading classification: i = internal, e = external, (f) = fixed, (o) = operable.

[h] Passive door ventilation with sound trap.

[i] Building operators are considering adding supplemental cooling to address added load from higher than designed occupant density.

## 5.2 Thermal comfort assessment

The thermal comfort assessment is based on qualitative feedback from occupant surveys, not calculated PMV-PPD.

**5.2.1 Compliance with ASHRAE 55.** The objective of ASHRAE Standard 55 is to have a "substantial majority (more than 80%) of the occupants" find their thermal environment "acceptable"; however, the advised method for verification is based on occupant survey asking about "satisfaction". We used this method to verify compliance to the standard, but we note that this shift from 'satisfaction' (in the question/scale used) to 'acceptability' (in the intent)

**Table 3. Temperature satisfaction by building.**

| Bldg. ID | # of occupant responses (response rate) | Percentage reported for temperature satisfaction | | |
|---|---|---|---|---|
| | | % satisfied considering votes from (-1) to (+3)[a] | % satisfied considering votes from (0) to (+3)[b] | % satisfied considering votes from (+1) to (+3)[c] |
| B1 | 78 (62%) | **89%**[e] | 67% | 63% |
| B2 | 28 (37%) | 64% | 39% | 32% |
| *B3*[d] | *23 (27%)* | *78%* | *61%* | *61%* |
| *B4*[d] | *47 (4%)* | **96%** | **89%** | *79%* |
| *B5*[d] | *41 (<1%)* | **93%** | **85%** | *73%* |
| B6 | 91 (48%) | 76% | 53% | 46% |
| B7 | 36 (53%) | **94%** | 75% | 64% |
| *B8*[d] | *207 (28%)* | **88%** | *72%* | *60%* |

[a] 'Slightly dissatisfied' (-1) is the lowest threshold for a positive vote for thermal acceptability in the ASHRAE 55–2017.

[b] 'Neither satisfied not dissatisfied' (0) is the lowest threshold for a positive vote for thermal acceptability in the ASHRAE 55–2013.

[c] The thermal comfort definition specifies a clear satisfaction statement.

[d] The buildings indicated in italic had a response rate lower than 35%.

[e] Bolden text for buildings that meets the ASHRAE 55 target of 80% satisfaction rate.

can be misleading. Furthermore, ASHRAE 55 modified its threshold for "acceptable" in the version ASHRAE 55–2017 [21], the standard suggests to include votes falling between '-1' ('slightly dissatisfied') and '+3' ('very satisfied'), while in the 2013 version [22], it asked to include votes between '0' ('neither satisfied not dissatisfied') and '+3' ('very satisfied').

The original dataset of 26 radiant buildings has 65% (17/26) of radiant buildings meeting the ASHRAE 55–2017 definition of acceptability [12], and 85% (22/26) meeting this definition if we consider 75% of occupant satisfied (instead of the ASHRAE threshold of 80%). Our subset of eight building is representative of the larger sample as 5/8 (62%) meet the ASHRAE 55–2017 definition of acceptability while 7/8 (85%) reach 75% of occupant satisfied for the same interval. Table 3 provides the results of the thermal comfort analysis considering all definitions. We bolded the text when the 80% criteria was met. We italicized the text for buildings which the response rate was less than the 35% (not recommended by the ASHRAE 55).

If we consider all eight radiant buildings in this dataset (independently from the response rate), three do not comply with any thermal comfort definitions, five buildings comply with ASHRAE 55–2017, two of which also comply with ASHRAE 55–2013, but no buildings were able to meet the thermal comfort definition based on satisfaction. If we only consider buildings that reached 35% response rate (B1, B2, B6, B7), two buildings comply with ASHRAE 55–2017, and no buildings were able to meet ASHRAE 55–2013 or the thermal comfort definition. The generally low compliance observed, despite the quality of the buildings analyzed, is aligned with the commonly observed low temperature satisfaction rate found in buildings [23]. Extending the interval to equate a negative response ('slightly dissatisfied') to a positive vote ('acceptable'), as in the 2017 version, is questionable jump in regard to what occupants reported about their conditions.

**5.2.2 Sources of satisfaction/dissatisfaction with thermal comfort.** Occupants that expressed dissatisfaction with temperature, were asked to select any or all of 20 listed sources of discomfort. Given that this is a "check all that apply" question and there are a different number of occupants per building, we represented the results in two ways: Fig 2(A) shows the percentage of dissatisfied votes *across all eight buildings* (n = 173), and Fig 2(B) shows the percentage of dissatisfied votes *per building*. This was done so that conclusions were not informed only by buildings with large occupancy and portion of dissatisfied occupants. We

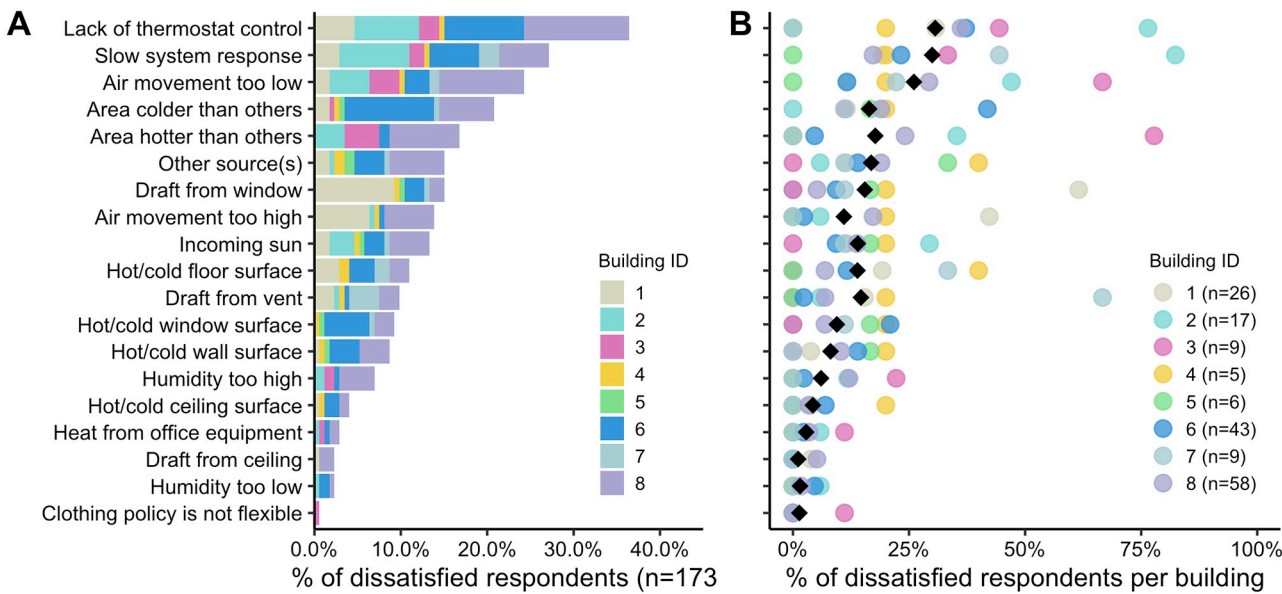

**Fig 2. (A) Percentage of dissatisfied occupants across all eight buildings (n = 173), and (B) percentage of dissatisfied occupants per building for each of the 19 potential sources of thermal discomfort (n by building).** The black diamond represents the average percent dissatisfied across each of the individual buildings.

considered two survey options as the same for these buildings: "thermostat is inaccessible" and "thermostat is controlled by other people". Only one building allowed occupants to make direct changes to the thermostat setpoints, so all responses are interpreted as "lack of thermostat control". An occupant's vote was only counted once if both options were selected. The survey is not designed to distinguish between system types, so occupant responses are inclusive of supplemental systems.

The order of sources of thermal discomfort in Fig 2(A) follows that observed for the 26 radiant buildings [12], suggesting that the eight buildings selected for this assessment are representative. Across all eight buildings, 173 of 551 occupants expressed dissatisfaction with temperature. As seen in Fig 2(B), there is variability in the top source of discomfort between buildings. This is not surprising given that each building is unique. Regardless of the variability in votes between buildings, there are clear trends in sources that are always or rarely selected, as suggested by the average percent dissatisfied across each of the individual buildings (black diamond).

Based on Fig 2, occupant open-ended responses, and building operator interviews, the following aspects appear to be related to thermal comfort in these buildings:

*The ability to quickly and individually change the thermal environment.* The top two sources of discomfort are "lack of thermostat control" and "slow system response". These results are not surprising given that TABS and ESS systems have high response times; if temperature setpoints are adjusted, it could take one hour up to several hours for those changes to be felt [16]. This is a primary reason why temperature setpoint changes is not a recommended control action to quickly address thermal discomfort in high thermal mass radiant system buildings. This is a concern regardless of the ability of radiant systems to be able to instantaneously extract part of the radiant load [24]. Additionally, the method used to modulate temperature setpoints and the placement of slab temperature sensors could further influence system response [25]. We do not have sufficient information to assess if radiant temperature control methods affected occupant comfort.

Satisfaction was higher in buildings where occupants were equipped with fast-response adaptive opportunities that enable either group control (i.e., operable windows, ceiling fans) or personalized control (i.e., desk fans, heaters) of thermal conditions. Three buildings had occupants indicate "my area is colder than others" and "my area is hotter than others" in the same open-plan office area of the building, exemplifying that individual occupants feel differently in the same environment. In such spaces where centrally controlled temperature setpoints cannot satisfy all occupants, building designers operators should consider offering individualized control (e.g., personal fans and heaters) to address individual thermal preferences and comfort.

"*I don't have much control over temperature. I usually run warm, so I like to have a fan.*"

"*I love the operable windows.*"

"*When it's too hot or cold it can take up to 2–3 days to be corrected. If you are in a fair bit of discomfort that is a long time to wait.*"

We note that loose dress codes are also supportive of individual comfort and provide an additional source of adaptive opportunity to occupants. Building operators for three buildings mentioned that adaptive opportunities supported energy goals by providing comfortable conditions during shoulder seasons without heating and cooling in the same day.

*User-controlled air movement.* Mechanically-supplied air in radiant buildings is generally for ventilation only and therefore at low velocity, but we know that increased air movement is preferred under neutral to warm temperature sensation [26]. There were 75% more complaints for "air movement too low" than for "air movement too high", suggesting occupants desire more air movement. The nature of the survey is to provide feedback on overall occupant experience, so we cannot correlate the desire for air movement with temperature or system operation; we can only conclude that occupants often feel the desire for more air movement. For occupants that indicated air movement was too high or felt discomfort from drafts, their comments revealed it was commonly due to automated (non-user controlled) features such as automated windows, trickle vents, and ceiling fans operating at too low temperatures, which is unpleasant [27]. Manually operable windows and desk fans appear to provide the best user-controlled air movement and building designers and operators should consider including these features or allowing occupant adjustment to ceiling fan operation.

"*The windows often open for airflow or for (what I assume) is anticipated higher temperatures later in the day, often leaving our space too cold.*"

(*Automated windows*)

"*Overhead fans in the past have gone on way too early and it seems to be too cool.*"

*Thermal comfort uniformity and overall temperature predictability.* Radiant design resources and researchers commonly reference uniform thermal conditions as an expected positive thermal comfort outcome in radiant buildings. There are multiple terms used to express this condition, including temperature uniformity and thermal comfort uniformity, and multiple cited outcomes and benefits, including 1) having a small vertical temperature gradient [4,7,28], 2) having a uniform spatial distribution of temperature [28], or 3) having uniform thermal comfort conditions (i.e., PMV) throughout a conditioned space [7,29]. In this study, as we did not conduct temperature measurements in the buildings, we only assessed the uniformity of thermal comfort conditions by relying on occupant subjective responses to the question "How satisfied are you with the temperature of your workspace?" on a 7-point scale and open-ended

responses. Occupant open-ended responses and building operator feedback indicate that there is uniform thermal comfort conditions, at least in open-plan office areas. Building operators reported that they receive fewer hot or cold spot complaints than in all-air buildings. Occupants in open-plan offices that selected "my area is colder than others" or "my area is hotter than others" typically referenced the space (or building) being too warm or too cool everywhere rather than in their particular workstation. Although this means that the thermal conditions were not considered comfortable, occupants rarely implied that spatial differences in temperature led to discomfort. Further, occupants suggest that there were predictable conditions throughout a space, throughout a day, or from day-to-day, which allows occupants to prepare accordingly. We hypothesize that these buildings have more stable interior temperatures due to the high thermal mass, but cannot confirm without physical measurements.

> "*The temperature is always fantastic, never too hot or too cold, there are no spots in the building where the temperatures vary significantly.*"

> "*It's often stuffy/hot in the morning in the summer, but I dress accordingly.*"

We are not drawing any conclusions on thermal comfort impacts from hot or cold surface or incoming solar radiation. As seen in Fig 2, only 11% of dissatisfied occupants identified floors as a source of discomfort and less than 10% identified hot or cold walls, windows, or ceilings. The low number of responses could be attributed to lack of occupant knowledge of radiant heat transfer rather than the absence of discomfort from these sources. Additionally, the building characteristic surveys indicated that buildings had well insulated envelopes and all had shading strategies to avoid direct solar heat gains through windows.

*Miscellaneous sources*. There are sources of comfort/discomfort that were unique to one or two buildings but could be relevant for other buildings outside this dataset. These include:

- Supplementary air-cooling systems in at least two buildings appear to be a cause of discomfort, including over-cooling in warm weather.

- Although spatial differences in temperatures did not appear to be a problem in open-plan offices, one building has zones that serve both open-plan office and private offices. Occupants in private offices more often responded that their space was "hotter than others" compared to those in the open-plan office, especially when the private offices did not have operable windows or a mechanism for air movement. It is common for temperature sensors to be located in open-plan offices in this scenario. Building designers should closely consider the thermal comfort impact of this type of design.

- Humidity levels were not identified as a problem in any of the buildings, but only one is located in a climate that experiences high outdoor relative humidity, with summer mean monthly wet bulb temperature around 23 ˚C.

- Operators in two buildings indicate that they make ad hoc and frequent changes to temperature setpoints in attempts to improve comfort, which is more akin to all-air system control. One building has large radiant zones (500–1000 m$^2$) with ESS and poor thermal comfort and complaints about inconsistent day-to-day temperatures, while the other building has small radiant zones (many less than 50 m$^2$) with TABS and thermal comfort is relatively good. We expect this type of operation to result in poor thermal comfort, as well as energy performance, due to the long system response times. However, the difference in thermal comfort results between the two buildings could be attributed to the zone size or system control factors that could be further investigated. Designers and operators should better understand

system control and could provide fast-response adaptive controls such as fans and heaters to occupants instead of making ad hoc changes to temperature setpoints.

We did not identify any single optimal radiant design or control strategy to maximize occupant comfort. However, the small sample size does not provide enough consistency in design and radiant system control across buildings to provide a reliable conclusion. The assessment does not indicate relationships between temperature satisfaction and the primary radiant surface, radiant loop control (e.g., variable/constant flow, variable/constant temperature), temperature setpoint strategy (e.g., zone air temperature setpoint, slab temperature setpoint, etc.), ventilation distribution, or how the system is operated outside of occupied hours, with the exception of the building in which operators make ad hoc changes to setpoints. This suggests that designing and operating TABS and ESS radiant systems to maximize energy efficiency shall not pose a significant threat to thermal comfort as long as design and operation are appropriate for the radiant system context. Additionally, we do not see any correlation between LEED certification and thermal comfort, providing further evidence that LEED certification is not strongly correlated with building performance [30].

**5.2.3 Feedback from building operators.** A benefit of radiant systems that has not been widely highlighted amongst the design community is improvements to building operation work load. Each of the six building operators had previous experience in traditional all-air buildings and all provided examples of how radiant systems positively impact their work. Their reasons include that the system is generally hands free, reduces the physical area of work to mostly the manifolds, which are outside of occupant areas, and has fewer mechanical parts for maintenance and repair.

All operators felt that radiant systems are more energy efficient compared to their experience with all-air systems, and they were generally pleased with the system's ability to provide comfort. However, some felt that they did not achieve as good of thermal comfort. Operators for two buildings stated that they have less granular and instantaneous control compared to all-air systems, and therefore, feel they lack the ability to address individual comfort, particularly in large zones covering open-plan office area. One of these buildings has poor thermal comfort and operators who make ad hoc changes to temperature setpoints for large zones, which is not a recommended operation strategy. In contrast, an operator in another building that makes frequent temperature changes with relatively small radiant zones felt that the more granular control was able to achieve acceptable thermal comfort, if not better than an all-air system. This design needs further investigation, as it is an unexpected finding.

**5.2.4 Energy performance and thermal comfort.** Although there was no single radiant design or operation that maximized thermal comfort within this small building sample size, we identified trends that promote both energy savings and thermal comfort. We were not able to assess the energy consumption of the radiant system by itself, only whole building consumption. Additionally, two buildings are campus-style and could not provide building-level energy data, and we were not successful in interviewing operators from the two highest energy consuming buildings. The following features appear to be related to energy performance and also promote thermal comfort in these buildings.

Of the four best energy performing buildings:

- All take advantage of free cooling through operable windows, or trickle vents in one building, which can improve thermal comfort from increased air movement in warm temperatures. At least one of these buildings turns off the radiant system operation to zones where windows are open, and one of these buildings relies solely on natural ventilation.

- All have zone air temperature deadband (i.e., degrees between heating and cooling air temperature setpoints) between 2.8 and 5.6 ˚C.

- Three use seasonal changeovers for the radiant system, which avoids heating and cooling in the same day. These buildings rely on operable windows, trickle vents, and/or personal control systems (i.e., desk fan, heaters) to maintain comfort during shoulder seasons.

- Multiple buildings have high performance envelopes, including sun shading to avoid direct solar heat gains, and reduce heat transfer.

  Of the two buildings with poor energy performance:

- Neither have operable windows for free cooling.

- At least one has a supplemental air-cooling system for hot and humid summer conditions that, based on occupant comments, appears to be overcooling the space. This building has the smallest dead band between heating and cooling (2.2 ˚C) and also has poor occupant comfort. This small dead band could be causing heating and cooling in the same day [31], and it could also be the cause of over-cooling in warm weather.

### 5.3 Acoustic quality assessment

**5.3.1 Percentage of occupants satisfied with acoustics.** In the IEQ survey used, satisfaction for acoustics is split between satisfaction with noise level and satisfaction with sound privacy. Noise level refers to general background noise, while sound privacy describes an occupant's ability to avoid being overheard in or overhearing other conversations. Although noise level and sound privacy are known sources of occupant dissatisfaction in buildings [18], there is no target guiding minimal occupant satisfaction with acoustic in spaces. In this section, we calculated acoustical satisfaction as the average between noise level and sound privacy per occupant and applied the ASHRAE 55 analysis process and thresholds described in Section 5.2.1 to arrive at the percentage of occupants finding the acoustics acceptable in the eight buildings. We also report the percentage of occupant satisfied with noise levels and with sound privacy. As seen in Table 4, both satisfaction with noise level and with sound privacy, indicated in parenthesis respectively, are generally low across all buildings, with sound privacy ranging

**Table 4. Acoustic satisfaction by building.**

| Bldg. ID | # of occupant responses (response rate) | Percentage reported for noise levels, sound privacy (in parenthesis) and acoustic satisfaction[a] | | |
|---|---|---|---|---|
| | | % satisfied considering votes from (-1) to (+3) | % satisfied considering votes from (0) to (+3) | % satisfied considering votes from (+1) to (+3) |
| B1 | 75 (60%) | (76%, 64%) \| 73% | (49%, 40%) \| 42% | (40%, 18%) \| 26% |
| B2 | 27 (36%) | (**82%**, 71%) \| 75% | (54%, 57%) \| 50% | (43%, 36%) \| 32% |
| B3[b] | 23 (27%) | (**83%**, 70%) \| 74% | (74%, 57%) \| 65% | (65%, 57%) \| 57% |
| B4[b,c] | 47 (4%) | (**89%**, 72%) \| **85%** | (**81%**, 51%) \| 66% | (70%, 45%) \| 47% |
| B5[b] | 41 (<1%) | (**85%**, 61%) \| 76% | (63%, 27%) \| 44% | (51%, 15%) \| 17% |
| B6 | 90 (47%) | (78%, 53%) \| 64% | (60%, 28%) \| 45% | (42%, 12%) \| 18% |
| B7 | 36 (53%) | (75%, 67%) \| 67% | (50%, 31%) \| 44% | (36%, 14%) \| 17% |
| B8[b] | 204 (27%) | (78%, 61%) \| 68% | (59%, 42%) \| 52% | (47%, 27%) \| 33% |

[a] We used the average between noise level and sound privacy per occupant to calculate satisfaction with acoustics per building.

[b] The buildings indicated in italic had a response rate lower than 35%.

[c] Bolden text is used when satisfaction rate meets the 80% threshold.

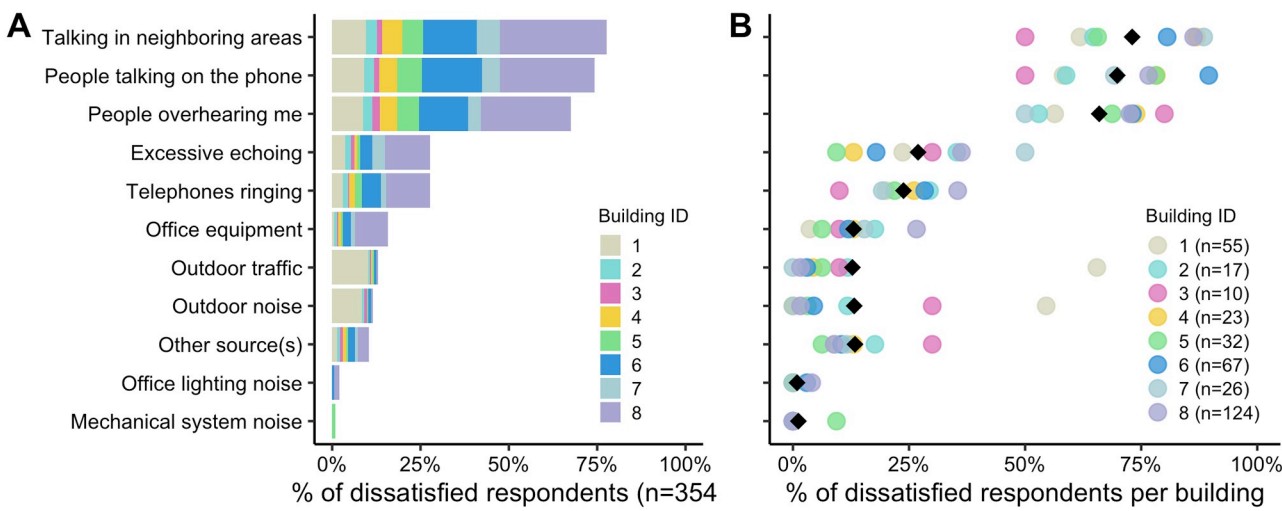

**Fig 3. (A) Percentage of dissatisfied occupants across all eight buildings (n = 354) and (B) percentage of dissatisfied occupants per building for each of the 19 potential sources of thermal discomfort (n by building).** The black diamond represents the average percent dissatisfied across each of the individual buildings.

from only 27% to 57% satisfied considering only neutral and positive votes. Only one building meets the 80% threshold for acoustics under the most lenient acceptability definition. Overall, these results show that occupants are even less satisfied with acoustics than they are with thermal comfort.

**5.3.2 Sources of satisfaction/dissatisfaction with acoustics.** Occupants that expressed dissatisfaction with acoustics (answering negatively to either noise level or sound privacy satisfaction), were asked to select any or all of 10 listed sources of discomfort. Given that this is a "check all that apply" question and there are a different number of occupants per building, we represented the results in two ways in Fig 3, similarly as what was done for the thermal comfort assessment.

Across all eight buildings, 354 of 543 occupants expressed dissatisfaction with noise and/or sound privacy, notably more than those that expressed dissatisfaction with temperature. 221 were dissatisfied with noise and 333 were dissatisfied with sound privacy; 200 were dissatisfied with both. The responses are in alignment with the quantitative survey study on 60 buildings, indicating the eight buildings are representative. Based on Fig 3, occupant open-ended responses, and building operator interviews, the following aspects appear to be related to acoustics in these buildings:

*Sound privacy in open-plan offices remains a challenge*. The top three causes of acoustical dissatisfaction in Fig 3 are more closely aligned with sound privacy than noise, as are the majority of open-ended responses. The primary space type in these buildings is open-plan office, which is detrimental to sound privacy. In current design practice, radiant systems push designs towards more open-plan space and the highly reflective thermally active surfaces for TABS and ESS systems can exacerbate the problem. However, there are other perhaps stronger factors driving designs towards open-plan (such as higher occupant densities, affordability, flexibility of the space) and therefore, we cannot attribute the cause to radiant systems alone.

"*Open office spaces need to have private areas both large and small for meetings or private conversations*"

"*I overhear technical conversations and my own interest in the technical issues is the problem. I end up listening to it instead of focusing on my own work.*"

*Exposed reflective surfaces may contribute to unwanted sound reverberation.* "Excessive echoing" and "telephones ringing" are the next most prevalent sources, but, notably, there is a large reduction in occupants selecting these as the source of their dissatisfaction. It is feasible that these sources could also be indirectly associated with the highly reflective surfaces like the following comments suggest; however, there are no acoustic pressure measurements to use as validation:

"*Lack of ceiling tile creates an echo chamber.*"

"*The building tends to echo quite a bit, I can hear people on first level all the way to the third level.*"

*Lack of noisy mechanical equipment.* As speculated by the design community, very few respondents identified mechanical equipment as an issue, which supports statements that radiant systems are quiet. However, 16% of occupants selected office equipment as a problem. In the building where this was primarily a problem, the issue appears to be two-fold: improperly sized ventilation diffusers that create a whistling noise and noisy ceiling fans, neither of which are directly related to the radiant system.

"*The mechanical heating and cooling system is very quiet.*"

*Few designs employ noise reduction strategies.* Six buildings have strategies in place to reduce noise issues in the studied buildings, including wall or vertical acoustic panels (two buildings), high partitions and a white noise generator (one building), carpeted floors (four buildings), and unique acoustically designed VanAir doors (one building). No buildings included horizontally hung acoustic clouds. One of the buildings that uses carpet on portions of the radiant floor as its only acoustic treatment has 51% of the occupants satisfied with acoustics, the second highest of all buildings. This building also has low occupant density, which could contribute to lower sound pressure levels in the space. The other solutions do not appear to be highly effective based on comments and satisfaction scores. Outdoor noises are primarily a problem in Building 3 with automated windows. There happened to be nearby construction at the time of the survey that could have influenced responses, so it is not conclusive that this would remain the primary source of dissatisfaction.

Acoustics continue to be a main area of design concern in buildings, much of it having to do with open-plan office and sound privacy. There have been few studies on whether radiant designs cause increased dissatisfaction, and successful studies will benefit from sound pressure measurements.

## 6 Limitations

Our analysis of the eight buildings is meant to provide insight into occupant satisfaction and perception that can help radiant building designers and building operators. Eight buildings represent a small sample, and we are aware that results may not be generalized. We chose buildings that showed various levels of occupant satisfaction and that were located in different climates to broaden the range of answers we could get. This also increased the variability between building designs and operation, which limited the common characteristics by which to assess.

Our analysis is based on information provided in the occupant, building characteristics, radiant design, and facility operation surveys. The building and radiant design surveys were

completed by knowledgeable contacts, and we assumed the information provided in these forms to be correct; we did not perform a fact checking review to assess the responses. The survey responses reflect the operation at the time they were filled and have limited ability to capture system-specific details. We did not gather any field measurements for factors influencing comfort or acoustics in the buildings. Additionally, we acknowledge that supplemental systems, such as upsized mechanical ventilation, may serve a portion of the heating and/or cooling loads in these buildings. More investigation is needed to better guide proper supplemental system sizing in high thermal mass radiant buildings.

The occupant satisfaction survey has pre-defined options, which may not capture nuances or could be interpreted differently. The survey is voluntary, and respondents are not required to answer every question, so survey completeness and response rate is a concern. ASHRAE 55–2017 guidance suggests 35% response rate to increase accuracy and representation of a building's population. We used occupant feedback from all of the eight buildings regardless of the response rate. Additionally, the occupant survey used in this analysis is meant to capture occupants' subjective perceptions of their typical experience in the space, not of specific episodic events (e.g., right-now survey).

## 7 Conclusion

We conducted a post-occupancy assessment in eight buildings using embedded radiant systems (TABS or ESS). We investigated over 500 occupant survey responses in all eight buildings and interviewed building operators in six. Five buildings had at least 80% occupants reaching the ASHRAE 55–2017 criteria of thermal acceptability and seven had at least 75% reaching this criterion. The primary factors leading to temperature discomfort in these buildings were the lack of control over the thermal environment, both temperature and air movement, and the slow system response (i.e., high response time) for these systems. Occupant comfort trends in these buildings were not unique to radiant buildings. Features that appear to resolve the comfort issues included fast-response adaptive opportunities, such as operable windows that allow for group control, and/or personal fans or heaters that allow for individual control based on user thermal sensation and preference. These are important for designers and operators to consider in radiant buildings due to slow response time of the systems. Other factors contributing to temperature satisfaction were low risk for unwanted air movement, likely due to lower airflow rates in radiant spaces compared to all-air, and predictable temperatures. We did not find a specific radiant system design or control scheme that clearly outperformed the other from the point of view of thermal comfort; although, the sample size is small. Acoustics had low satisfaction across all eight buildings, and most issues stem from sound privacy in open-plan offices; there was no strong evidence linking sources of acoustic dissatisfaction with the radiant design. Strategies such as carpets and acoustical panels should be further explored for effectiveness, especially in open-plan office spaces.

## Supporting information

**S1 Dataset. Aggregated occupant satisfaction responses.**
(XLS)

## Acknowledgments

The authors would like to thank all the designers, engineers, building operators, building managers, facility managers, and other building stakeholders who collaborated with us, provided building information, and made this study possible.

## Author Contributions

**Conceptualization:** Megan Dawe, Stefano Schiavon, Fred Bauman.

**Data curation:** Megan Dawe, Caroline Karmann, Fred Bauman.

**Formal analysis:** Megan Dawe, Caroline Karmann.

**Funding acquisition:** Stefano Schiavon, Fred Bauman.

**Investigation:** Megan Dawe, Caroline Karmann.

**Methodology:** Megan Dawe, Fred Bauman.

**Project administration:** Fred Bauman.

**Supervision:** Stefano Schiavon, Fred Bauman.

**Visualization:** Megan Dawe.

**Writing – original draft:** Megan Dawe, Caroline Karmann.

**Writing – review & editing:** Stefano Schiavon, Fred Bauman.

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
