## [Decision Letter · Decision Letter 0]

7 Jun 2021

PONE-D-21-01672

Field evaluation of thermal and acoustical comfort in eight North-American buildings using embedded radiant systems

PLOS ONE

Dear Dr. Dawe,

Thank you for submitting your manuscript to PLOS ONE. After careful consideration, we feel that it has merit but does not fully meet PLOS ONE’s publication criteria as it currently stands. Therefore, we invite you to submit a revised version of the manuscript that addresses the points raised during the review process.

We look forward to receiving your revised manuscript.

Kind regards,

Forrest Meggers

Academic Editor

PLOS ONE

Journal Requirements:

2. Thank you for including your ethics statement: "The occupant survey data was collected and analyzed anonymously. IRB approval was granted for the surveys under CPHS protocol 2010-05-1550. Under this approval, only staff at the research institute are able to access and use the survey data."

7. Thank you for submitting the above manuscript to PLOS ONE. During our internal evaluation of the manuscript, we found significant text overlap between your submission and the following previously published work, of which you are an author.

https://escholarship.org/uc/item/6d95z6sw

Please revise the manuscript to rephrase the duplicated text, cite your sources, and provide details as to how the current manuscript advances on previous work. Please note that further consideration is dependent on the submission of a manuscript that addresses these concerns about the overlap in text with published work.

Additional Editor Comments:

Dear Megan, Caroline, Stefano, and Fred,

First, I apologize for the delay and thank you for the excellent article and patience. I had a third reviewer I had hoped would complete a review, but I'm proceeding without it. The reviewers suggested revisions, one minor and one major. I am proceeding with suggesting minor revisions, but while I don't think the article needs major changes, it does need some major expansion in certain aspects detailed in the comments from the reviewers. One aspect I like about PLOS ONE is that the reviewer have the option to unblind their identity, which Reviewer 2 has done, and as Eric suggested major revisions feel free to work directly with him on his detailed comments in the attached PDF. I think both reviewers point to some issues with a lack of clarity between heating and cooling scenarios. I agree the relationship between draft and fans and in heating vs cooling scenarios should be considered how the body convectively exchanges with air that is either closer or farther from skin temperature is - Both of the broad and specific comments #3 from Eric and also mentioned by Reviewer 2 in the context of controls.

I am happy to provide additional feedback if needed, and I promise to very quickly turn around your revised submission, and again sorry for the delay. It has been a challenging year and I hope you have all stayed safe and healthy.

best,

/Forrest

Reviewers' comments:

Reviewer's Responses to Questions

**Comments to the Author**

1. Is the manuscript technically sound, and do the data support the conclusions?

Reviewer #1: Yes

Reviewer #2: Yes

2. Has the statistical analysis been performed appropriately and rigorously? 

Reviewer #1: Yes

Reviewer #2: Yes

3. Have the authors made all data underlying the findings in their manuscript fully available?

Reviewer #1: Yes

Reviewer #2: Yes

4. Is the manuscript presented in an intelligible fashion and written in standard English?

Reviewer #1: Yes

Reviewer #2: Yes

5. Review Comments to the Author

Reviewer #1: See attachment. My main concerns for this paper are in reference to novelty in the interpretation of a small dataset in a field increasingly driven by large datasets for informed decision making. I believe there is a lot of novel potential, and a few reforming operations will greatly benefit this piece.

Reviewer #2: Review

In section 4.1, you should add a caveat to the use of TABS.

Compared to radiant panels, TABS has a slow response time while panels are the fastest responding system. This is a significant difference. Another critical difference is that TABS has challenging control and basically, setpoints don't exist. The reason is that temperature measured across the slab thickness does not reflect slab surface temperature, especially when covered with carpet.

In section 4 or section 5 or the Limitation section, you need to recognize the limitation of the data collection method. This is a widely used data collection method, but it has an inherent error that results can be biased, especially when we have low response rates. Lassen et al. (2020) reported this phenomenon. Discussion of this issue required a separate paper, so I am only asking to recognize this limitation, hoping that the scientific community will improve data collection or better understand the results. For example, we will understand what it means to have 30 responses out of 300 occupants, who is responding or voting and why. You address other issues very well in section 5.

Lassen, N., Goia, F., Schiavon, S. and Pantelic, J., 2020. Field investigations of a smiley-face polling station for recording occupant satisfaction with indoor climate. Building and Environment, 185, p.107266.

In Section 4.3 please add categories used. For example, how did you categorize information you received about the radiant system design?

Please also include some indication of heating and cooling operation. How many buildings used both vs only one of these options.

Line 276 - response time of the TABS is the essential factor. Another factor is the temperature of the surface that is exchanging heat with the occupants. How do we modulate that temperature? Slab temperature, slab surface temperature, and carpet (or similar top covering of the slab exchanging heat with the occupant) can have significantly different temperatures. You can find more details in Pantelic et al., (2018). Please discuss this issue also.

Pantelic, J., Schiavon, S., Ning, B., Burdakis, E., Raftery, P. and Bauman, F., 2018. Full scale laboratory experiment on the cooling capacity of a radiant floor system. Energy and Buildings, 170, pp.134-144.

Line 350 - how many design strategies you evaluated? In Table 2 there is a summary of the systems, but DOAS means that airside supplies only outdoor air, and there is no information about the size of air vs. radiant system. Radiant loop control would be equivalent to coil water side control that we never really consider in any discussion on thermal comfort, so I think this is a wrong parallel. As I mentioned above, the critical problem besides slow response is what is setpoint controlling? Perhaps this is a good place to discuss this since you mentioned other design issues.

Lines 375-380 - this is interesting information. I understand that you can't make a significant conclusion based on the information provided, but it certainly points out an issue that needs further investigation.

In the Conclusion section, you mention that one of the comfort improvement methods is to use a personal fan. That is true for warm conditions in summer. What would be the method for winter operation and cold conditions?

I don't see support for the sentence "We did not find a specific radiant system design or control scheme that clearly outperformed the other from the point of view of thermal comfort." In my opinion, you didn't evaluate the system design, so please remove this sentence from the Conclusion section.

6. PLOS authors have the option to publish the peer review history of their article (what does this mean?). If published, this will include your full peer review and any attached files.

Reviewer #1: **Yes: **Eric Teitelbaum

Reviewer #2: No

---

## [Author Response · Author response to Decision Letter 0]

20 Jul 2021

We appreciate the thorough review and comments from reviewers and the editor. We have attached detailed responses to each reviewer comment and a new letter to the editor describing our revisions.

---

## [Decision Letter · Decision Letter 1]

8 Oct 2021

Field evaluation of thermal and acoustical comfort in eight North-American buildings using embedded radiant systems

PONE-D-21-01672R1

Dear Dr. Dawe,

We’re pleased to inform you that your manuscript has been judged scientifically suitable for publication and will be formally accepted for publication once it meets all outstanding technical requirements.

Kind regards,

Forrest Meggers

Academic Editor

PLOS ONE

Additional Editor Comments (optional):

First, thank you for your patience. Finding good reviewers has been a challenge.

Congratulations on the excellent paper. I do agree with the reviewers initial suggestions for more analysis, but I also appreciate your response and recognize that more work is forthcoming. The new Reviewer 3 does bring up some excellent points, and while I'm suggesting an accept now, I hope you will consider adding a couple sentences in your discussion to address issues of sizing air side and radiant side components in the types of HVAC systems you have analyzed. As reviewer 3 points out, by not addressing system size aspects you leave a very open avenue for critique. I think you can get ahead of those criticisms by mentioning the air-side sizing and by adding a couple comments on the implications, as suggested.

Thanks again for the hard work putting this together and please reach out if you would like any additional feedback.

Reviewers' comments:

Reviewer's Responses to Questions

**Comments to the Author**

1. If the authors have adequately addressed your comments raised in a previous round of review and you feel that this manuscript is now acceptable for publication, you may indicate that here to bypass the “Comments to the Author” section, enter your conflict of interest statement in the “Confidential to Editor” section, and submit your "Accept" recommendation.

Reviewer #1: All comments have been addressed

Reviewer #3: All comments have been addressed

2. Is the manuscript technically sound, and do the data support the conclusions?

Reviewer #1: Yes

Reviewer #3: Yes

3. Has the statistical analysis been performed appropriately and rigorously? 

Reviewer #1: Yes

Reviewer #3: Yes

4. Have the authors made all data underlying the findings in their manuscript fully available?

Reviewer #1: Yes

Reviewer #3: Yes

5. Is the manuscript presented in an intelligible fashion and written in standard English?

Reviewer #1: Yes

Reviewer #3: Yes

6. Review Comments to the Author

Reviewer #1: Thank you for taking my comments into such thorough consideration, even the very high level ones. It is quite a fine manuscript, looking forward to seeing the final, formatted work.

Reviewer #3: Thank you for this article. I want to be clear that I am not one of your original reviewers, but I have assesssed the reviewer comments, and agree that comments have been mostly addressed as identified.

I do have one point to raise, that was partially raised by different reviewers on the generalisability of findings and the lack of thorough analysis of the design history of these buildings.

I would like to offer an analogy to an issue I have with this assessment. In the general discourse of HVAC assessment of real-world buildings, we know that 'oversizing' of HVAC systems is a historical / persistent problem in industry. It is, by any measure, a common problem (see Djuneady et al. 2011). It's been observed and discussed so sufficiently in research literature, that we accept it as a norm. Any field study of a random sample of commercial buildings designed to ASHRAE standards is likely to encounter oversized HVAC systems.

I propose that the same must apply to building with radiant systems, and here we have to accept how an oversized embedded radiant system would perform. If I combine a TABS with a DOAS, but I oversize that DOAS (and it has a heating / cooling coil), you can bet that DOAS will begin to do heavy lifting with respect to regulating indoor air and humidity. So it brings up the question, is an embedded radiant system with an oversized ventilation system a proper radiant system, or is it just a radiant-assisted mechanical ventilation system. The ERI@N office of NTU in Singapore comes to mind, which was precisely this: a radiant system with so much mechanical ventilation provided that the sensible cooling provided by the ceiling was trivial.

Anecdotally, for those of us in this field, we know that many radiant systems have not been implemented in practise in a way where the radiant system is truly the primary medium of heating and cooling delivery. Once I start seeing a building with several 'supplementary' mechanical ventilation systems, I get an innate feeling that these are buildings where the radiant system - though advertised as important - ultimately becomes ancillary to the ventilation system comfort-wise.

My question to you is how do you discern this, and can you? I propose two changes to the paper: to expand in the discussion on this issue, and provide some discussion on the extent to which the ventilation systems are satisfying the bulk of the sensible heating and cooling needs in these spaces.

Ideally, I would recommend that you provide more data on these buildings. What are the sizes of the DOAS systems implemented? How much air are they providing? What kind of setpoints are the DOAS systems using for heating and cooling? And yes, ultimately, are the ventilation systems contributing significantly or marginally to sensible heating and cooling indoors? Without acknowledging this, as an expert in radiant systems, I think this reads as a weak study because it avoids acknowledging this very issue that is at the heart of radiant system design (and the success failure of historical projects).

7. PLOS authors have the option to publish the peer review history of their article (what does this mean?). If published, this will include your full peer review and any attached files.

Reviewer #1: **Yes: **Eric Teitelbaum

Reviewer #3: No

---

## [Editor Report · Acceptance letter]

18 Oct 2021

PONE-D-21-01672R1 

Field evaluation of thermal and acoustical comfort in eight North-American buildings using embedded radiant systems 

Dear Dr. Dawe:

I'm pleased to inform you that your manuscript has been deemed suitable for publication in PLOS ONE. Congratulations! Your manuscript is now with our production department. 

Kind regards, 

on behalf of

Dr. Forrest Meggers 

Academic Editor

PLOS ONE